# Pathway elucidation of bioactive rhamnosylated ginsenosides in *Panax ginseng* and their de novo high-level production by engineered *Saccharomyces cerevisiae*

Chaojing Li[1,2,5], Xing Yan[1,5], Zhenzhen Xu[1,3,5], Yan Wang[1], Xiao Shen[1], Lei Zhang[4], Zhihua Zhou [1✉] & Pingping Wang [1✉]

Rg2 and Re are both rhamnose-containing ginsenosides isolated exclusively from *Panax* plants, which exhibit broad spectrum of pharmacological activities. However, limitations of current plant-relied manufacturing methods have largely hampered their medical applications. Here, we report elucidation of the complete biosynthetic pathway of these two ginsenosides by the identification of a rhamnosyltransferase PgURT94 from *Panax ginseng*. We then achieve de novo bio-production of Rg2 and Re from glucose by reconstituting their biosynthetic pathways in yeast. Through stepwise strain engineering and fed-batch fermentation, the maximum yield of Rg2 and Re reach 1.3 and 3.6 g/L, respectively. Our work completes the identification of the last missing enzyme for Rg2 and Re biosynthesis and achieves their high-level production by engineered yeasts. Once scaled, this microbial biosynthesis platform will enable a robust and stable supply of Rg2 and Re and facilitate their food and medical applications.

[1] CAS-Key Laboratory of Synthetic Biology, CAS Center for Excellence in Molecular Plant Sciences, Chinese Academy of Sciences, Shanghai, China. [2] University of Chinese Academy of Sciences, Beijing, China. [3] School of Life Sciences, Henan University, Kaifeng, China. [4] Logic Informatics Co., Ltd., Shanghai, China. [5] These authors contributed equally: Chaojing Li, Xing Yan, Zhenzhen Xu. ✉email: zhouzhihua@cemps.ac.cn; ppwang@cemps.ac.cn

Rg2 and Re are protopanaxatriol (PPT)-type ginsenosides containing a rhamnose moiety isolated exclusively from *Panax* plants[1,2]. In dried *P. ginseng* root, Rg2 and Re are primarily distributed in the root hair at concentrations of approximately 0.6 and 2.2 mg/g, respectively[1] (Supplementary Table 1). According to *Chinese Pharmacopoeia*, Re content is one of the standard criteria for ginseng quality. Pharmacological studies have demonstrated that Rg2 exhibited therapeutic activity in the prevention of lipopolysaccharide (LPS)-induced acute inflammation[3], offered protection against UV-B radiation-induced tissue damage, and ameliorated high-fat diet-induced metabolic disease[4,5]. Moreover, many studies have indicated that Rg2 has potent neuroprotective effects, making it a promising candidate drug for the treatment of memory impairment and neuronal death[6]. Pharmacological studies have also demonstrated that Re may protect the cardiovascular system[7]. Indeed, Re is effective against isoproterenol-induced myocardial fibrosis and heart failure[8], and exerts anti-ischemic and anti-arrhythmic effects in mice.

Currently, Rg2 and Re are extracted from *Panax* plants, however, the cultivation of *Panax* plants is heavily relied on the use of pesticide and chemical fertilizer, which may cause environmental pollution and influence the quality of ginsenoside. Besides, the fields for *Panax* plant cultivation generally require a period of 7-15 years for rotational tillage before the next replantation[9]. In addition, purification of such compounds with low content from plant materials is not economic and may bring resource waste.

Synthetic biology offers an attractive alternative for large-scale, high-quality production of natural plant products. The complete biosynthetic pathways of protopanaxadiol (PPD) and PPT, the skeleton for most ginsenosides, have been resolved since 2011. In this process, PgDDS catalyzes the first committed step during the production of dammarenediol-II (DM) from 2, 3-oxidosqualene[10,11]. Cytochrome P450s CYP716A47 and CYP716A53v2 then hydroxylate C12 of DM and C6 of PPD to produce PPD and PPT, respectively[12,13]. Since then, the downstream glycosylation pathways of many ginsenosides have been uncovered, and microbial cell factories to produce these ginsenosides have been constructed. For instance, the complete biosynthetic pathway of ginsenoside CK was elucidated and introduced into *Saccharomyces cerevisiae* chassis in 2014[14]. The de novo production of CK from glucose was accomplished in yeast at a very low titer (~1.4 mg/L)[14]. After systematic optimization of precursor biosynthesis and the UDP-glucose concentration in CK-producing yeast-cell factories, the highest production of CK was recently reported at a titer of 5.7 g/L[15]. Recently ginsenosides Rh2[16–20], Rg3[16,17], Rh1[21], F1[21], and Rg1[21], and notoginsenosides R2 and R1[22,23], have also been produced using engineered yeast-cell factories. All these reports clearly indicate that synthetic biology offered a powerful way for high-level and sustainable ginsenoside production. However, one challenge preventing the production of Rg2 and Re by such way has been the lack of a completely characterized biosynthetic pathway, in which an unknown UDP-glycosyltransferase (UGT) was deduced to responsible for the C6-O-Glc rhamnosylation of Rh1 or Rg1.

Rhamnosylation is a glycosylation modification seen in natural products, especially among flavonoids, terpenoids, and lipids. Many natural rhamnosylated plant products exhibit attractive biological activities, such as myricetin-3-O-β-rhamnoside, which is found in the leaves of *Myrtus communishas* and may facilitate wound healing and anti-hepatitis B virus effects[24,25]. Quercetin-3-rhamnoside is another natural product that can inhibit influenza A virus replication[26]. Although many rhamnosylated natural products have been isolated, the rhamnosyltransferases responsible for the biosynthesis of these compounds remains largely unknown. Indeed, less than 30 rhamnosyltransferases have been reported, most of which are involved in the biosynthesis of flavonoids. AtUGT78D1 from *A. thaliana*, the first reported rhamnosyltransferase of natural products, catalyzed the transfer of rhamnose from UDP-rhamnose to the 3-OH position of quercetin and kaempferol to yield quercetin 3-O-rhamnoside and kaempferol 3-O-rhamnoside, respectively[27]. AtUGT89C1 converts kaempferol 3-O-glucoside into kaempferol 3-O-glucoside-7-O-rhamnoside[28], and DzGT1 from *Dioscorea zingiberensi* rhamnosylates triterpenoid trillin to form prosapogenin A[29].

In the present study, we sought to develop a high-titer microbial fermentation-based platform for the production of Rg2 and Re. We combined gene discovery and functional assays to characterize a UGT94-familiy gene (*PgURT94*) from *P. ginseng* and completely resolve the biosynthetic pathway of Rg2 and Re. Based on a previously constructed PPT- and Rg1-producing yeast chassis[23], the complete biosynthetic pathway of Rg2 and Re was reconstructed in yeast. De novo high-level production of Rg2 and Re was achieved through strain engineering and fed-batch fermentation. This work also establishes a successful example of the high-level production of rare natural plant products, especially those with rhamnosylation modifications.

## Results and discussion

**Discovery of the missing UDP-glycosyltransferase for ginsenoside Rg2 and Re biosynthesis**. Previously, we systematically characterized a series of UGTs involved in the biosynthesis of ginsenosides and completely resolved the biosynthetic pathway of Rh1 and Rg1[21]. The downstream pathway from Rh1 and Rg1 to Rg2 and Re is speculated to be catalyzed by an unknown UGT enzyme (Fig. 1). Because all previous characterized UGTs responsible for the sugar elongation of ginsenosides belong to the UGT94 family[30], we thus focused our efforts on screening UGT candidates belong to this family.

We identified 665 UGTs with >350 amino acids residues (typical plant UGTs' length) from the *P. ginseng* transcriptome that were predicted to have a conserved Plant Secondary Products Glycosyltransferase (PSPG) box. These could be clustered into 187 OTUs with a 95% cutoff. Twenty-two OTUs and their representative UGTs which possessed >40% amino acid identity to the previous identified UGT94 family UGT member, PgUGT94Q2[30] were recognized as belong to the UGT94 family. Through gene co-expression analysis of these UGT candidates, previously characterized P450s, and UGTs involved in PPT-type ginsenoside biosynthesis from *P. ginseng* transcriptome data, a UGT94 gene named *PgURT94* was identified. The expression pattern of *PgURT94* was strongly correlated with *PgDDS*, *CYP716A47*, *CYP716A53v2*, *PgUGT71A53*, and *PgUGT71A54*, which are involved in Rh1 and Rg1 biosynthesis (Fig. 2a). Besides, *PgURT94* was highly expressed in the root hairs of *P. ginseng*, which is consistent with the distribution of Re in this tissue (Supplementary Table 1). Thus, we speculated that PgURT94 is most likely to be involved in the biosynthesis of Rg2 and Re.

To verify this hypothesis, we firstly cloned this gene from callus of *P. ginseng*. The *PgURT94* gene has an open reading frame (ORF) of 1404 bp, encoding a protein of 467 amino acids. PgURT94 protein has a sequence identify of 46.1%, 47.1%, and 44.8% with PgUGT94Q2 (UGT catalyzing C3-O-Glc glucosylation of PPD-type ginsenosides), PgUGT94Q3 (UGT catalyzing C6-O-Glc glucosylation of PPT-type ginsenosides), and PgUGT94Q6 (UGT catalyzing C20-O-Glc glucosylation of PPD and PPT-type ginsenosides), respectively[30]. For the enzymatic activity test, PgURT94 was initially expressed in *E. coli* (Supplementary Fig. 1) and the crude enzymes from *E. coli*

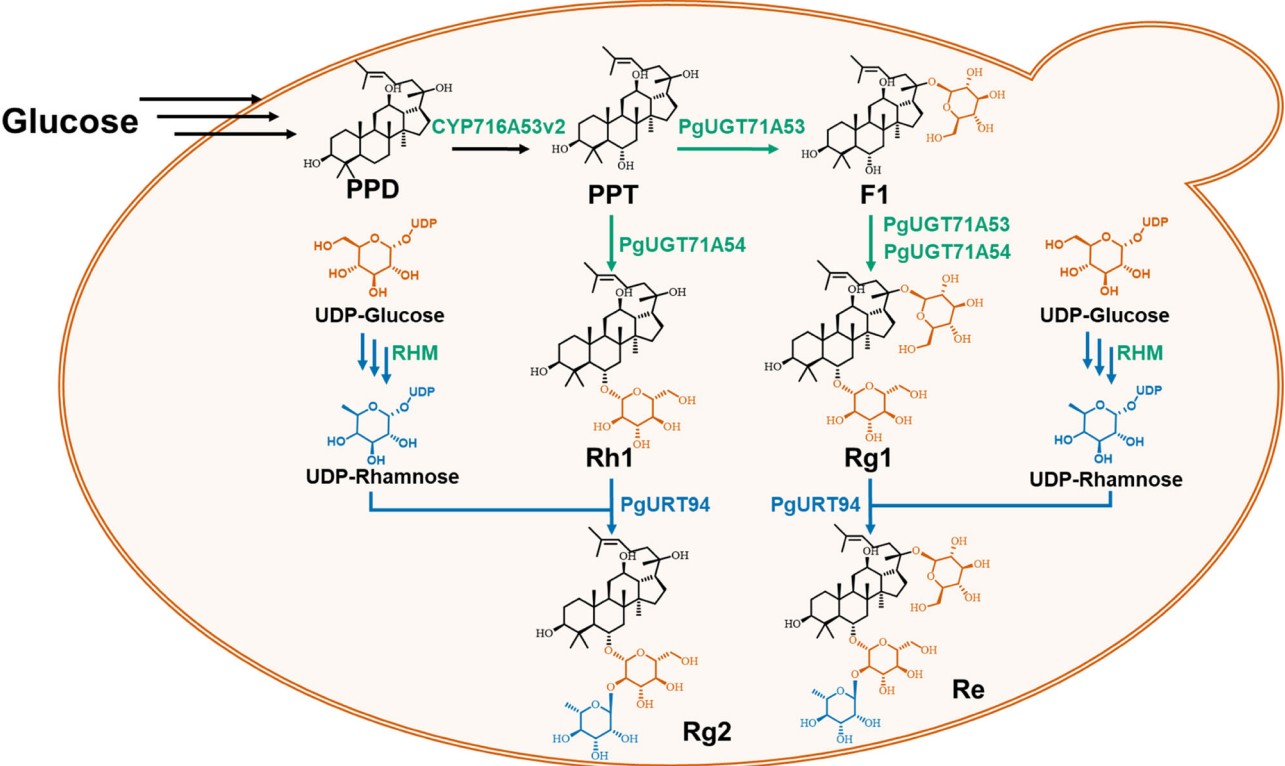

**Fig. 1 The proposed pathway for de novo biosynthesis of ginsenosides Rg2 and Re in engineered yeast strains.** Bluish green arrows represent glycosylation steps using UDP-glucose as a sugar donor, blue arrows represent the biosynthetic pathway of UDP-rhamnose and glycosylation steps using UDP-rhamnose as a sugar donor. Multi-step conversions were presented as multi arrows. Bluish green marked genes represented previous reported genes and the bule marked gene represented the identified one in this study. PPD protopanaxadiol, PPT protopanaxatriol.

expressing PgURT94 were incubated with Rh1 and Rg1 as substrates, and UDP-Rha as a sugar donor. The reaction products were subjected to HPLC analysis and results indicated a product was generated in the reaction extract from PgURT94 and Rh1 incubations, which had the same retention time as the Rg2 standard. This compound was not detected in the control reaction with Rh1 and crude enzyme of E. coli strain harboring empty pET28a vector (Fig. 2b). A product was also observed in the reaction extract from PgURT94 using Rg1 as a substrate, and was monitored along with the Re standard (Fig. 2c). The structures of these two newly produced compounds were confirmed to be Rg2 and Re, respectively, by HPLC/electrospray ionization mass spectrometry (ESIMS) (Supplementary Fig. 2) and NMR (Supplementary Fig. 3).

To test the sugar donor specificity of PgURT94, in vitro enzymatic assays were performed by using UDP-glucose as a sugar donor and incubating PgURT94 with Rh1 and Rg1, respectively. To ensure accuracy of the assay, a previously reported UGT (PgUGT94Q3) which could catalyze the glycosylation modification of Rh1 and Rg1 using UDP-glucose as a sugar donor[30], was used as a positive control. TLC and HPLC analyses of the reaction extracts revealed that while production of glycosylated products Rf and C20-O-Glc-Rf could be detected by PgUGT94Q3 as expected, no products were detected by PgURT94, indicating that PgURT94 could not use UDP-glucose as a sugar donor (Supplementary Fig. 4). These results demonstrated that PgURT94 is a specific rhamnosylation UGT.

**Reconstruction and optimization of the ginsenoside Rg2 biosynthetic pathway in Saccharomyces cerevisiae.** Through functional characterization of PgURT94, the complete biosynthetic pathway of Rg2 became clear: PgUGT71A54 catalyzes the C6-OH

glycosylation of PPT to form Rh1, and PgURT94 then transfers a rhamnose moiety to the C6-O-Glc of Rh1 to produce Rg2 (Fig. 1). To achieve de novo biosynthesis of Rg2 in yeast, codon-optimized PgUGT71A54 and PgURT94 (hereafter referred to as synPgURT94), under the control of two strong constitutive promoters respectively, were introduced into the chromosome of strain PPT-10, a PPT-producing chassis constructed in our previous work[23]. Since S. cerevisiae lacks the native UDP-Rha biosynthetic pathway, AtRHM2 from A. thaliana, which catalyze the formation of UDP-Rha from UDP-glucose, was also expressed in PPT-10[31,32]. The resulting strain, called Rg2-01, produced 36.8 mg/L Rg2 according to the analysis of metabolites in subsequent flask fermentations (Fig. 3a, b).

The total triterpenoid production (PPD + PPT + Rh1 + Rg2) in strain Rg2-01 decreased sharply compared to that of the parent strain PPT-10 (Fig. 3b). Since the triterpenoid biosynthetic pathway is a highly NADPH-consuming pathway and the biosynthesis of UDP-Rha from UDP-Glc by AtRHM2 in the Rg2-producing strain Rg2-01 is also an NADPH-dependent pathway[33], we thus focused our attention on the NADPH consumption. For PPT biosynthesis, the precursor pathway from acetyl-CoA to 2, 3-oxidosuqalene requires three NADPH molecules (the formation of mevalonate from 3-hydroxy-3-methylglutaryl-CoA need two NADPH molecules[34] and the formation of 2, 3-oxidosqalene from squalene need one NADPH molecule[35]). Besides, two P450s that catalyze the formation of PPT from DM also consume two NADPH molecules[12,13] (Supplementary Fig. 5). Therefore, the NADPH supply is of great importance for PPT production. We speculate that the introduced of NADPH-dependent UDP-Rha synthase AtRHM2 may further increase NADPH consumption and caused a reduction in total triterpenoid production in strain Rg2-01 (Supplementary Table 6).

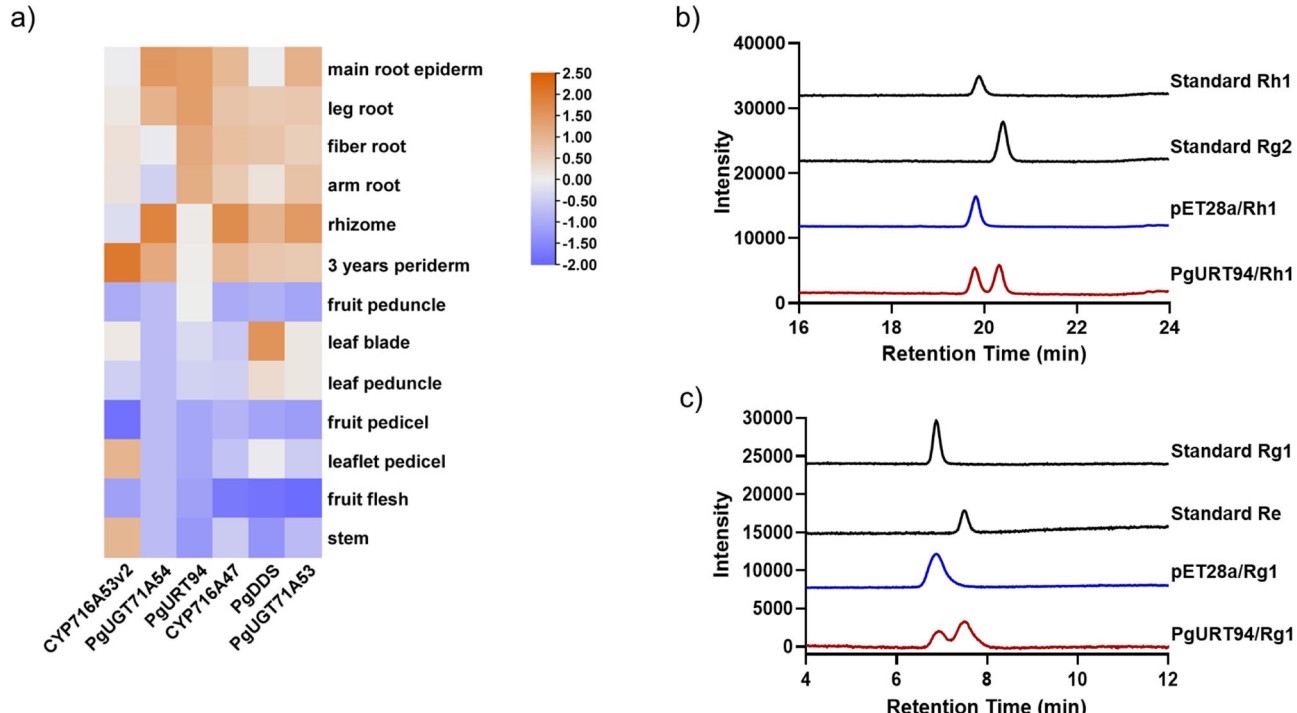

**Fig. 2 Expression pattern analysis of PgURT94 and its enzymatic activity toward Rh1 and Rg1 as substrates. a** Heat-map analysis of the relative abundance of *PgURT94* expression, along with *PgDDS*, *CYP716A47*, *CYP716A53v2*, and *PgUGT71A53* in different parts of *P. ginseng*. **b** HPLC analysis of the in vitro reaction products catalyzed by PgURT94 crude enzyme using Rh1 as sugar acceptor and UDP-rhamnose as sugar donor. **c** HPLC analysis of the in vitro reaction products catalyzed by PgURT94 crude enzyme using Rg1 as sugar acceptor and UDP-rhamnose as sugar donor. Crude enzymes of *E. coli* strain harboring pET28a empty vector were used as a negative control for above assays and authentic ginsenoside samples Rh1, Rg1, Rg1, and Re were monitored as standards.

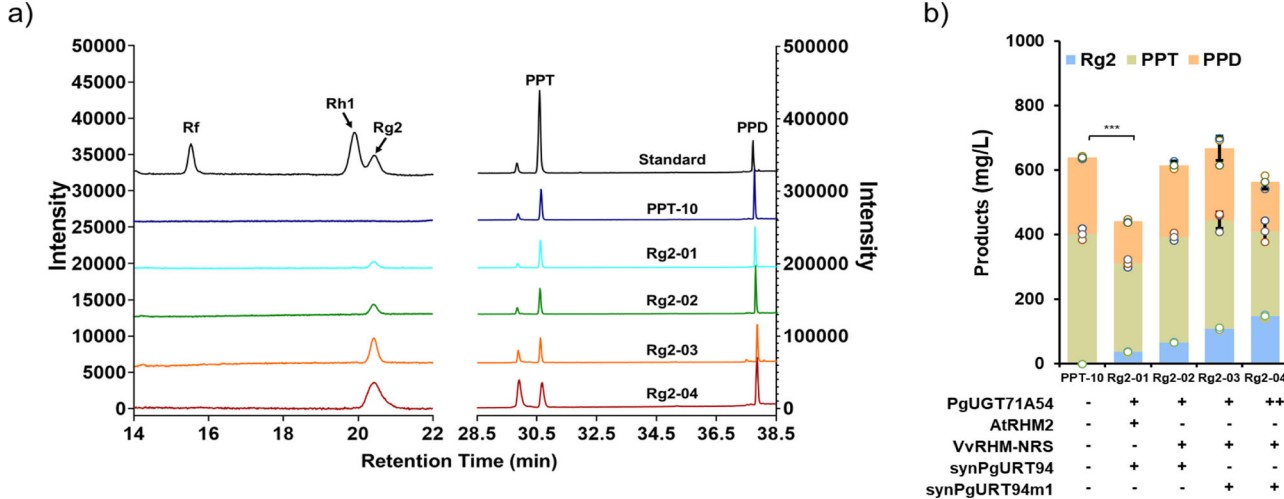

**Fig. 3 Analysis of the ginsenoside Rg2 production in a engineered *Saccharomyces cerevisiae* strain. a** HPLC analyses of Rg2, Rh1, PPT and PPD production in yeast strains Rg2-01, Rg2-02, Rg2-03, and Rg2-04. The PPT chassis strain PPT-10 was used as a control. Mixed samples of Rf, Rh1, Rg2, PPT and PPD were monitored as standards. **b** Quantitative analysis of Rg2 and its related intermediates Rh1, PPT, and PPD in yeast strains Rg2-01, Rg2-02, Rg2-03, and Rg2-04. Genetic modification of each strain was drawn under the column, "+" represent the strain possess the corresponding engineering, while "−" represent the corresponding engineering are missing the strain. All data represent the mean of *n* = 3 biologically independent samples and error bars show standard deviation.

To address this issue, we utilized a NADPH-independent UDP-Rha synthase to alleviate the NADPH limitation. Several studies demonstrated that an engineered RHM enzyme (VvRHM-NRS) formed by fusing a bifunctional UDP-4-keto-6-deoxy-d-glucose 3,5-epimerase (NRS)/UDP-4-keto-rhamnose 4-keto-reductase (ER) from *A. thaliana* to the N-terminal of a *Vitis* *vinifera* UDP-Rha synthase VvRHM can be a self-sufficient NADPH-independent enzyme for UDP-Rha synthesis[36,37]. *VvRHM-NRS* and *synPgURT94* were then introduced into PPT-10 to construct strain Rg2-02. Metabolite analysis of strain Rg2-02 indicated that Rg2 production increase to 66.4 mg/L, which is approximately 1.8-fold the amount of Rg2-01 (Fig. 3b). No

significant reduction in total triterpenoid production was observed compared to the parent strain, PPT-10. These results clearly demonstrate that utilization of a NADPH-independent enzyme for UDP-Rha synthesis could rescue the triterpenoid reduction resulting from insufficient NADPH supply.

**Boosting ginsenoside Rg2 production by the utilization of a mutated synPgURT94 and the enhancing of PgUGT71A54 expression level**. During screening the single clones of the construction of strain Rg2-02, a clone, hereafter designated as strain Rg2-03, was found to produce Rg2 with a production of 107.5 mg/L, which is significantly higher than Rg2-02. The *synPgURT94* gene of Rg2-03 was then amplified and sequenced; a missense T-to-A mutation of the 163th nucleotide was found, which resulted in a leucine to methionine mutation at the 55th amino acid. To explore whether this amino acid mutation contributed to the enhanced production of Rg2 in strain Rg2-03, we expressed the synPgURT94 mutant (synPgURT94m1) in *E. coli* and performed an in vitro enzymatic activity assay. Using Rh1 as a substrate, the conversion ratio of Rh1 to Rg2 by synPgURT94 and synPgURT94m1 was 70.6% and 92.4%, respectively, which demonstrated a great improvement in catalysis efficiency by synPgURT94m1 (Fig. 4a, c, Supplementary Table 7). The improved catalytic performance of synPgURT94m1 may explain the increased Rg2 production of strain Rg2-03. The catalytic activity of synPgURT94m1 towards Rg1, to produce Re, was also assessed; significant enhancement of catalytic activity was also observed (Fig. 4b, d, Supplementary Table 7). Therefore, this mutant was used for Re-cell factory construction.

Since Rh1 did not accumulate in any of the engineered Rg2-producing strains, the conversion of PPT to Rh1 may be a limiting step in Rg2 production. To address this, we introduced an additional copy of *PgUGT71A54* into strain Rg2-03 to create Rg2-04. As expected, the production of Rg2 by Rg2-04 reached 147.1 mg/L, representing a 40% improvement compared to Rg2-03 (Fig. 3, Supplementary Table 6). By combining all engineering strategies, the production of Rg2 increased 4.0-fold from Rg2-01 to Rg2-04. However, in our final strain, there was still more than 263.3 mg/L of PPT accumulated and no Rh1 was detected (Supplementary Table 6). Therefore, the conversion of PPT into Rh1 remains a major bottleneck for Rg2 production. We believe that the production of Rg2 could be further improved by addressing this limiting step in the future.

**Construction of a yeast-cell factory for the production of ginsenoside Re**. A Rg1-producing yeast cell factory, Rg1-02, was constructed previously by inserting *PgUGT71A53* and *PgUGT71A54* respectively into the single-copy *YORWΔ22* and multi-copy *delta DNA* sites. The production level of Rg1 was 111.45 mg/L in shake flasks and 1.95 g/L in fed-batch fermentations[23]. For Re production in yeast, *VvRHM-NRS* and *synPgURT94m1*, under the control of strong constitutive promoters, were introduced into Rg1-02 to generate strain Re-01 (Fig. 1). Subsequent shaken flask fermentation tests detected Re at a production level of 215 mg/L (Fig. 5b, Supplementary Fig. 6).

Re production in strain Re-01 was much higher than Rg2 production in strain Rg2-04, despite the fact that Re has a more complicated biosynthetic pathway. The Re content is also much higher than that of Rg2 in *Panax* plants, including *P. ginseng* and *P. notoginseng*[1,2]. To test whether this phenomenon is determined by some intrinsic factors or just a coincidence, we then examined the difference between the two yeast strains. Re-01 and Rg2-04 share the same PPT-producing background strain and downstream rhamnosylation pathway (Fig. 1), thus PgUGT71A53 and

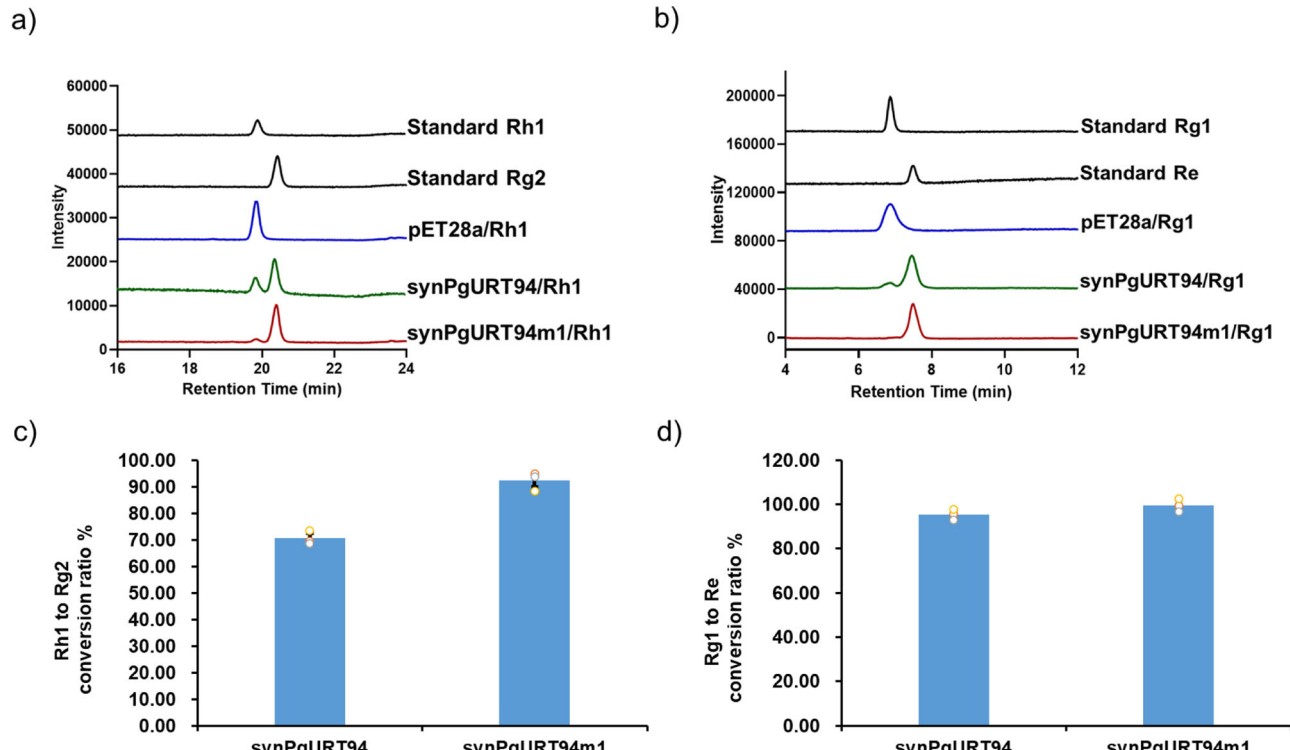

**Fig. 4 Enzymatic assay of synPgURT94 and its mutant toward Rh1 or Rg1 as substrate. a**, **b** HPLC analyses of the in vitro reaction products catalyzed by synPgURT94 and synPgURT94m1 crude enzymes using Rh1 (**a**) and Rg1 (**b**) as the sugar acceptor and UDP-rhamnose as the sugar donor. Crude enzymes of *E. coli* strain harboring pET28a empty vector were used as a negative control and authentic ginsenoside samples Rh1, Rg1, Rg1 and Re were monitored as standards. **c**, **d** Quantitative analysis of the conversion ratio catalyzed by synPgURT94 and synPgURT94m1, from Rh1 to Rg2 (**c**) and from Rg1 to Re (**d**). All data represent the mean of $n = 3$ biologically independent samples and error bars show standard deviation.

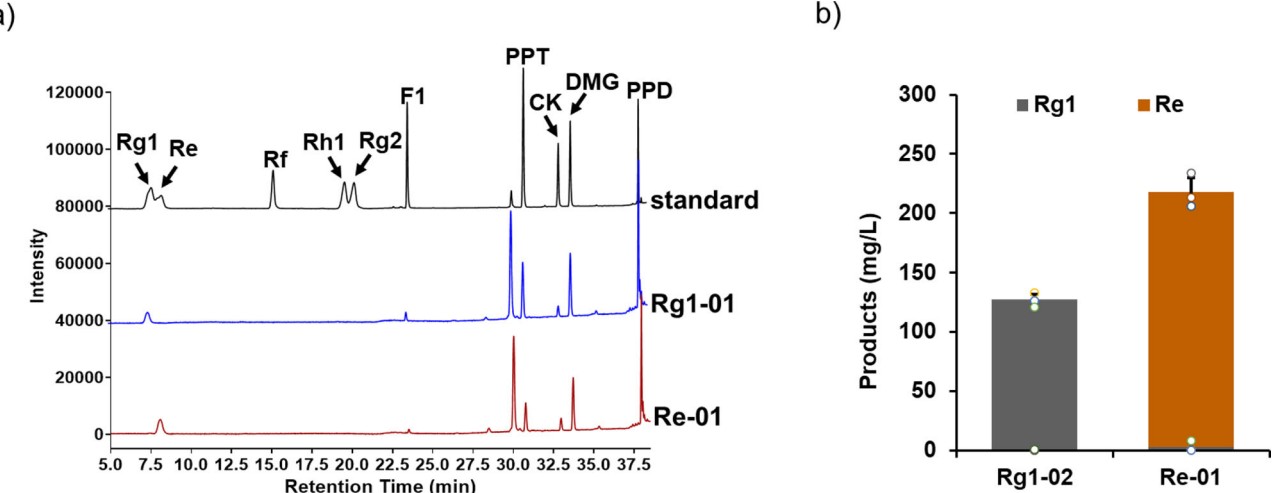

**Fig. 5 Analysis of the ginsenoside Re production in engineered *Saccharomyces cerevisiae* strains. a** HPLC analyses of Rg1, Re, F1, PPT, CK, DMG, and PPD produced in yeast strain Re-01. Mixed samples of ginsenosides were monitored as standards. The Rg1 chassis strain Rg1-02 was used as a negative control. **b** Quantitative analysis of Re and its related intermediate Rg1 in engineered yeast strains. All data represent the mean of $n = 3$ biologically independent samples and error bars show standard deviation.

PgUGT71A54 may contribute to different production between the two strains.

Previous enzyme kinetics assays indicated that the $k_{cat}/K_m$ value of PgUGT71A53 for PPT is higher than that of PgUGT71A54 by more than 400-fold[23]. Thus, the poor PPT-catalyzing efficiency of PgUGT71A54 in strain Rg2-04 may severely limit the formation of Rh1 and lead to the lower production of Rg2. Accordingly, the high efficiency of PgUGT71A53 towards PPT ensured the rapid conversion of PPT into F1 which can be subsequently converted into Rg1 by PgUGT71A54. To assess the catalytic performance of PgUGT71A54 towards F1, enzyme kinetics assays were performed. Although the $k_{cat}/K_m$ value of PgUGT71A54 towards PPT is low ($2.58 \times 10^{-2}\,\mathrm{mM^{-1}\,s^{-1}}$), its $k_{cat}/K_m$ value towards F1 is much higher ($7.87 \times 10^{-1}\,\mathrm{mM^{-1}\,s^{-1}}$) (Supplementary Table 8). We also observed that PgUGT71A53 could also catalyze the conversion of F1 to Rg1, with a $k_{cat}/K_m$ value of $7.40 \times 10^{-2}\,\mathrm{mM^{-1}\,s^{-1}}$. Thus, the elevated activity of PgUGT71A54 and PgUGT71A53 towards F1 enabled highly efficient conversion of F1 into Rg1 in the Re-01 strain. These results demonstrated that the elaborate coordination of PgUGT71A53 and PgUGT71A54 in the pathway precisely regulated the production of downstream products. The enzyme characteristic of UGTs might determine the contents of Re and Rg2 in *Panax* plants, as well as that of our engineered yeast strains.

**High-titer production of ginsenoside Rg2 and Re by fed-batch fermentation**. With the above engineering efforts, we obtained two yeast strains (Rg2-04 and Re-01) that could in turn produce ginsenosides Rg2 and Re directly from glucose, respectively. However, the production titer in shaken flasks remained at relatively low levels. We previously reported a series of successful examples of the promotion of ginsenosides through fed-batch fermentations in bioreactors, including Rh2, CK, Rg1, NgR1, and NgR2, all of which reached the gram-per-liter scale after optimization of the fed-batch fermentation conditions (Supplementary Table 9). To achieve higher production of Rg2 and Re, we performed high-density fed-batch fermentation of Rg2-04 and Re-01 using a 1.3-L parallel bioreactor system. Since Rg2-04 and Re-01 have the same strain background (strain PPT-10) to previous constructed NgR1- and NgR2-producing strains, the

fermentation control parameters were set as reported previously[23]. Fresh medium was fed into the fermenter at approximately 24 h and the cell biomass of both strains continuously increased after feeding. For strain Rg2-04, the OD600 continuously increased to a maximum of 330.4 at 108 h and remained unchanged until 120 h. Re-01 reached a growth plateau at 96 h (OD600 = 456.8) and exhibited a slight decrease at 108 h (Fig. 6 and Supplementary Table 6). Unexpectedly, the final cell biomass of Re-01 is more than 37.2% higher than that of Rg2-04, although the mechanism underlying this phenomenon is unclear.

The final production amount of Rg2 by Rg2-04 was 1.3 g/L, which represents an 8.9-fold increase over shaken flask production. Other important ginsenosides precursors, including PPD and PPT, could also be detected in the fermentation broth, with titers of 2.2 and 3.0 g/L, respectively (Fig. 6, Supplementary Table 6). The final production level of Re by Re-01 reached 3.6 g/L, which represents a 16.6-fold increase over shaken flasks. Many important ginsenoside precursors, such as PPD, DMG (20S–O-β-(D-glucosyl)-dammarenediol-II), CK, PPT, F1, and Rg1, could also be detected in the fermentation broth, with titers of 1.0, 0.5, 0.7, 0.2, 0.1, and 0.2 g/L, respectively (Fig. 6 and Supplementary Table 6). The accumulation of numerous triterpenoid precursors in both strains indicated that some rate-limiting steps remained in the engineered strains, and there is great potential for even higher Rg2 and Re production levels if these obstacles can be overcome. Since the physicochemical properties of these ginsenoside precursors vary significantly, it will not be difficult to separate and purify them from the fermentation broth. Thus, engineered strains Rg2-04 and Re-01 may also be useful for the preparation of these valuable ginsenosides.

## Conclusions

We discovered and characterized a rhamnosyltransferase PgURT94 from *P. ginseng*, which catalyzes C6-O-Glc rhamnosylation of Rh1 and Rg1 to produce Rg2 and Re, respectively, and thus resolved the complete biosynthetic pathway of Rg2 and Re. By reconstructing their complete biosynthetic pathway in yeast and addressing the major limiting steps, we accomplished high-level de novo production of these two ginsenosides from glucose using engineered yeast. With both Rg2 and Re production reaching a titer of more than 1 g/L, industrial, large-scale production of these valuable ginsenosides through microbial

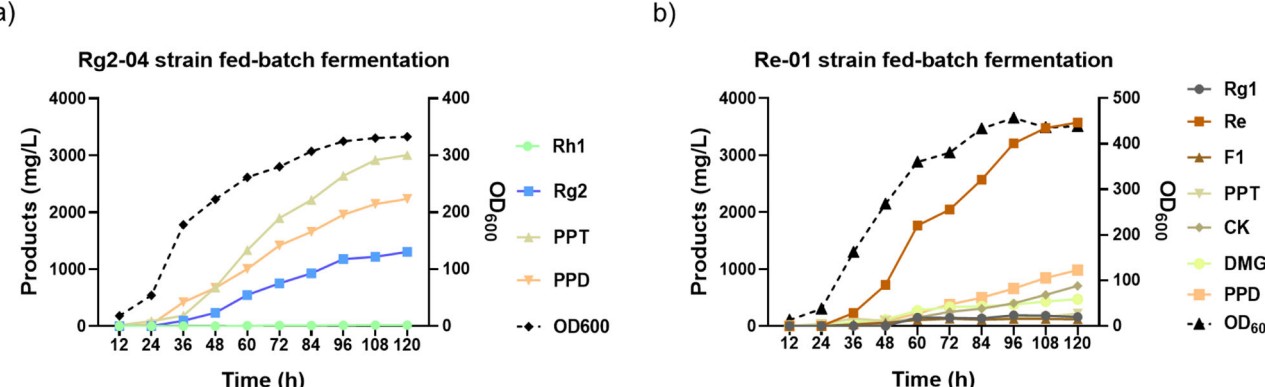

**Fig. 6 Time course analysis of yeast strain growth and ginsenosides Rg2 and Re production in fed-batch fermentation. a** Time course analysis of cell growth and triterpenoid production of strain Rg2-04. **b** Time course analysis of cell growth and triterpenoid production of strain Re-01. *PPD* protopanaxadiol, *PPT* protopanaxatriol, *DMG* 20*S*–O-*β*-(D-glucosyl)-dammarenediol-II.

fermentation is expected in the near future. Beyond elucidating the complete pathway of Rg2 and Re production, our work represents a major step towards the industrial synthesis of Rg2 and Re by constructed yeast-cell factories that can increase production at the gram-per-liter scale.

## Method

**Chemicals, plasmids and strains**. Authentic ginsenoside standards protopanaxadiol (PPD), protopanaxatriol (PPT), compound K, ginsenoside F1, Rh1, Rg2, Rf, Rg1, and Re were purchased from Nantong Feiyu Biological Technology (Jiangsu, China). Plasmids pMD18-T (TaKaRa, Dalian, China) and pET28a (Merck, Germany) were used for UGTs cloning and expressing, respectively. *E. coli* strain TOP10 was used for gene cloning, and BL21 (*DE3*) was used for UGTs heterologous expression. Plasmids pUG66, pAG25 and pSH69 obtained from EUROSCARF were used as the template for the amplification of *ble*, *NatMX* and *hygroB* selection marker, respectively. *S. cerevisiae* strain PPT-10 and Rg1-02[23] constructed in previous study were used as the parent strain for all engineering. The genes *PgURT94*, *PgUGT71A54* (Genbank accession No. KP795113.1), *PgUGT71A53* (KF377585.1), *PgUGT94Q3*[30], *AtRHM2* (Q9LPG6.1), and *VvRHM-NRS*[36] were codon-optimized and synthesized by Genscript Corporation (Nanjing, China).

**Predicting, cloning and heterologous expression of UDP-glycosyltransferases**. We obtained the raw data of transcriptome sequencing data and genome assembly data from NCBI SRA database (Supplementary Tables 10 and 11). FASTQC (version 0.11.9) was used to assess the quality of the raw data, and Trimmomatic (version 0.39) was used to remove the connector and end low quality sequences. Detailed method for the transcriptome assembly was described in supplementary information (Supplementary Methods). ORFs encoding UDP-glycosyltransferases were identified by the conserved plant secondary product glycosyltransferase (PSPG) box.

The *PgURT94* gene was amplified from *P. ginseng* root hair using primers listed in Supplementary Table 2 and cloned into the plasmid pMD18-T. For heterologous expression, *PgURT94* was then inserted into the *Nco*I and *Sal*I sites of pET-28a vector with a C-terminal 6 × his-tag. The resulting plasmid pET28a-PgURT94 was then transformed into *E. coli* BL21 (*DE3*) to generate strain pET28a-PgURT94-BL21(*DE3*). Expression of PgURT94 was performed by first cultivation this strain to a log phase (~2 h), followed by adding 0.2 mM IPTG and cultivation for additional 18 h at 16 °C. The cells were collected by centrifugation and suspended in 100 mM phosphate buffer (pH 7.0) and then lysed by homogenization method using French Press (26 kpsi). After centrifugation (12,000 g, 4 °C, 10 min), the cell debris was removed and the supernatant was used for enzymatic assay and western blot analysis[38]. All primers and strains used for cloning and heterologous expression of UGTs are listed in Supplementary Table 2.

**In vitro enzymatic assays**. Enzymatic assays for PgURT94 were carried out in a 100 μL reaction system containing 100 mM phosphate buffer (pH 7.0), 1% Tween-20, 5 mM UDP-rhamnose, 0.5 mM substrate (Rh1 or Rg1), and 80 μL of UGT crude enzyme. Reaction was carried out by incubating in water bath for 2 h in a 35 °C. Reaction products were extracted by adding 100 μL of *n*-butanol and vortexed for 2 h. After centrifugation (12,000 g, 10 min), the organic phase was then evaporated and dissolved in 100 μL methanol for Thin-layer chromatography (TLC) and High-performance liquid chromatography (HPLC) analysis. For sugar donor specificity assay of PgURT94, in vitro reactions were conducted similarly by using UDP-glucose as sugar donor instead of UDP-rhamnose. A previous reported

glucotransferase PgUGT94Q3 of Rh1 and Rg1 was used as a positive control. Reaction products were analyzed by both TLC and HPLC.

For the enzyme kinetics assays of PgUGT71A53 and PgUGT71A54, the UGT protein concentration in the crude enzymes were quantified by dot blot hybridization[39,40] using anti- his-tag antibodies (Yeasen Biotechnology, Shanghai, China). The purified UGT OleD (ABA42119.2) with C-terminal 6× His-tagged was diluted serially to make the standard curve for quantification. For the kinetics analysis of UGTs toward sugar acceptor (F1), the 500 μL reaction mixture containing 100 mM phosphate buffer (pH 7.0), 1% Tween-20, 5 mM UDP-glucose, 75–600 μM ginsenoside F1, and 100 μL crude enzyme (~60 ng/μL) were incubated at 35 °C for 50 min. The products were extracted by adding 500 μL of *n*-butanol. The organic phase was evaporated and dissolved in methanol for HPLC quantitative analysis. Lineweaver-Burk plot was used to calculate the Michaelis–Menten parameters. All data are presented as means ± SD of three independent repeats.

**Construction of yeast strains**. The previously constructed yeast strains PPT-10 and Rg1-02 were used as start strain for the construction of Rg2- and Re-producing strains in this study[23]. Generally, each target gene paired with a promoter and terminator was PCR amplified, and then co-transformed into yeast together with a selection marker and homologous recombination arms using standard ssDNA/LiAc yeast transformation methods. Because each adjacent fragment was designed to share over 60-bp identical sequence, they will be joined together via yeast homologous recombination and integrated into target chromosome site. For instance, strain Rg2-01 was constructed by transforming genes *AtRHM2*, *PgUGT71A54*, and *PgURT94* into the *delta DNA* site, a multi-copy site of PPT-10. The above three genes, their respective promoters (*TEF2p*, *TEF1p*, *UAS-TDH3p*), terminators (*TDH2t*, *CYC1t*, *FBA1t*), the yeast selection marker *ble* as well as two homologous arms of *delta DNA* site were amplified by PCR using primers listed in Supplementary Table 4. Then, these PCR fragments were purified by Gel Recovery kit (Magen Biotech, Shanghai, China) and then quantification by Nano-300 (Micro-spectrophotometer). Finally, ~1 μg of each PCR fragment was mixed together and co-transformed into PPT-10 to generated strain Rg2-01. Other strains were constructed by the similar way. All the *S. cerevisiae* strains used or constructed in this research are listed in Supplementary Table 3, the primers and genes used for the construction of the strains are listed in Supplementary Table 4 and Supplementary Table 5, respectively.

**Yeast strain cultivation and metabolites extraction**. Yeast strains were cultured in YPD medium (10 g/L Bacto Yeast Extract, 20 g/L Bacto peptone and 20 g/L dextrose) or synthetic medium (8 g/L KH$_2$PO$_4$, 15 g/L (NH$_4$)$_2$SO$_4$, 5.9 g/L succinate, 5 g/L lysine, 19.5 g/L dextrose, 6.2 g/L MgSO$_4$, Vitamin solution and Trace metal solution)[41]. For yeast transformants selection, 300 mg/L Hygromycin B (Sangon Biotech, Shanghai, China), 200 mg/L Nourseothricin (Sangon Biotech, Shanghai, China), 200 mg/L Phleomycin (Yeasen Biotechnology, Shanghai, China) were added into the solid YPD medium when appropriate. For the shake flask fermentation, individual yeast clones were inoculated into 5 mL synthetic medium in a test tube and cultivated at 30 °C, 250 rpm for 16 h to make the seed culture. Then 100 μL seed culture was inoculated into 10 mL of synthetic medium in 50 mL shake flasks and grown at 30 °C, 250 rpm for 4 days. For metabolites analysis, 500 μL of the fermentation broth was extracted with equal volume of *n*-butanol adequately, and the upper organic phase was evaporated and dissolved in methanol for HPLC analysis. Fed-batch fermentation of strain Rg2-04 and Re-01 were conducted in 1.3-L parallel bioreactor system using synthetic medium. The seed cultures were prepared by cultivating Rg2-04 or Re-01 individual clone into 50 mL synthetic medium at 30 °C, 250 rpm for 24 h. About 30 mL seed cultures were inoculated into 270 mL batch medium for fermentation. Temperature and pH of

the fermentation was maintained as 30 °C and 5.0, respectively. Dissolved $O_2$ was controlled above 30% by varying speed of agitation or air flow rate[19]. Feeding rate was controlled to keep ethanol concentration lower than 0.5 g/L.

**Chemical analysis**. TLC analysis was performed using TLC silica gel 60 $F_{254}$ plates (Meck, Germany) with chloroform/methanol/water (120/10/1, v/v/v) as the developing solvent. The spots on the TLC plates were visualized by spraying with 1% (w/v) vanillin in $H_2SO_4$/ethanol (6/100, v/v) followed by heating at 110 °C for 3 min. For identification, ginsenoside authentic samples, Rh1, Rf, Rg1 and C20-O-Glc-Rf (dissolved in ethanol) was also spotted on the plate.

HPLC analysis was performed on a Shimadzu LC20ADXR system (Shimadzu, Kyoto, Japan) equipped with a pumper, an auto-sampler and a diode array detector. Chromatographic separation of PPD, PPT, DMG, and ginsenosides compound K, F1, Rf, Rh1, Rg2, Rg1, and Re was carried out at 35 °C on a Boltimate C18 column (100 mm × 2.1 mm, 2.7 μm, Welch, Shanghai, China). The gradient elution system consisted of water (A) and acetonitrile (B). Separation was achieved using the following gradient: 0–7 min (17.5% B), 7–10 min (17.5%–24% B), 10–20 min (24% B), 20–21 min (24%–32.5% B), 21–25 min (32.5% B), 25–35 min (32.5%–65% B), 35–36 min (65%–95% B), 36–39 min (95% B), 39–40 min (95%–17.5% B), 40–43 min (17.5% B), and the flow rate was kept at 0.45 mL/min. Ginsenoside samples were detected at 203 nm. For the conformation of chemical structures, mass spectrometry[42] and nuclear magnetic resonance (NMR) analyses[43] were employed and described in Supplementary Materials (Supplementary Methods).

**Statistics and reproducibility**. Statistics data were analyzed by GraphPad Prism 7.0 or TBtools software. The results were presented as bar graphs with mean of $n = 3$ biologically independent samples and error bars show standard deviation, or provided in tables as mean ± SD. Student's t test was used for the comparison of data. $p$ value < 0.001 was considered as statistically significant and marked with three asterisks.

**Reporting summary**. Further information on research design is available in the Nature Research Reporting Summary linked to this article.

## Data availability

All data generated in the study are provided in the article or Supporting Information file, other relevant data that support the findings of this study are available from the corresponding author upon reasonable request. The uncropped western blot image of Supplementary Fig. 1 was shown in Supplementary Fig. 7. Gene sequences of synPgURT94 and synPgURT94m1 were deposited in RDBSB (the Registry and Database of Bioparts for Synthetic Biology, https://www.biosino.org/rdbsb) under accession Nos. of OENC366046 and OENC366047, respectively.

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

## Acknowledgements

This work was financially supported by the National Key Research and Development Program of China (Grant No. 2018YFA0900700), the National Natural Science Foundation of China (Nos. 32071425; 31921006), the Strategic Priority Research Program of the Chinese Academy of Sciences (Grant No. XDB27020206), the Strategic Biological Resources Service Network Plan of the Chinese Academy of Sciences (Grant No. KFJ-BRP-009; KFJ-BRP-017-60), the National Key Research and Development Program of Yunnan Province (2019ZF011-1), and the Tianjin Synthetic Biotechnology Innovation Capacity Improvement Project (Grant No. TSBICIP-KJGG-002-17).

## Author contributions

P.W. and Z.Z led the project. C.L., X.Y., Z.X., P.W., and Z.Z were responsible for this study concept and design. C.L. and Z.X. performed the methodology, data curation, validation. C.L., X.Y., and P.W. performed the writing—original draft, writing—review & editing. Y.W. investigated this study. X.S. investigated this study. L.Z. investigated this study. Z.Z. performed the writing—original draft, writing—review & editing, and project administration.

## Competing interests

The authors declare no competing interests.
