## [Peer Review File · Communications Biology]

Reviewers' comments:

Reviewer #1 (Remarks to the Author):

The authors have successfully attempted to engineer the bio synthetic pathway for production of rhamnosylated ginsenosides in a yeast chassis based production module in a high titre.

Indeed, information related to the rhamnosyl transferases particularly were extremely limited and this report adds tremendously to our present understanding of the biosynthetic pathway of the panaxosides. The work is technically sound and should be easily reproducible. The narrative is well written, concise and is easy to interpret. The results are well explained and readily comprehensible.

I recommend the manuscript to be accepted in the present form. However I would like the authors to comment on the ginsenoside extraction protocol that they have employed from fermentation broth. The ginsenosides were extracted directly from the organic upper layer of n butanol and analyzed further. However, this may also result in loss of ginsenosides during the phase separation. Have the authors tested for ginsenoside recovery using this extraction method and estimated a percentage recovery? Since extraction of ginsenosides vary with solvents and methanol is almost universally used for ginsenoside extraction. In case of broths, they may be lyophilized and then extracted using methanol and then phase separation using n butanol may be done. Please refer to the following for more clarity on my query.

Biswas T, Ajayakumar PV, Mathur AK, Mathur A. Solvent-based extraction optimisation for efficient ultrasonication-assisted ginsenoside recovery from *Panax quinquefolius* and *P. sikkimensis* cell suspension lines. *Natural Product Research*. 2015 Jul 3;29(13):1256-63.

I would like to congratulate the authors for this successful attempt to unravel the complex biosynthetic pathways for bioactive panaxosides and opening up avenues for their large scale production via synthetic biology.

Reviewer #2 (Remarks to the Author):

Ginsenosides showed various pharmacological activities that have been wide used for preventing cardiovascular and cerebrovascular diseases. Rg2 and Re are main ginsenosides with sugar moiety in *Panax* plants root which take 3-4 years to harvest. The manuscript elucidated the biosynthesis pathway of these two ginsenosides and high yield production in engineered *Saccharomyces cerevisiae* with synthetic biology strategy. The products and yield were proved by TLC, NMR and LCMS which were persuasive enough to me. This work is well fundamental build and potential to produce large-scale ginsenosides in days.

In Suppl. Fig.5. The chemistry structures need to be redrawn.

Reviewer #3 (Remarks to the Author):

The manuscript describes continuing research of the authors on the rhamnosylated ginsenosides pathway in *Panax ginseng* and their production by engineered *Saccharomyces cerevisiae*. The study is promising, and the paper contains some interesting results of potential scientific significance. However, there is major insufficiency that the investigators need to address in the organization and presentation of the paper. It must be revised in a logical flow and clear manner in order to further bring out the objectives and importance of the various experiments. The figure legends are not properly explained, it should be in detail. Need consistency in the use of gene, vector/strain names (e.g. PPT, pET28a-RH1, pgURT94-Rh1), and repetition of the same information e.g. introduction: lines 61-61, lines 82-83, and lines 96-98 convey the same message.

The methodology part needs to improve with more clarity: substrate and its concentration used and other details for each experiment should be mentioned.

Figure 4 shows only the peaks, which are already shown in the figures before. Better to bring some meaningful data from supplementary.

In figure 2 a and figure 4a authors find peaks for rh1 but in other places they did not find it.

Table S7 and lines 192-193 indicate 70.6% and 92.4% conversion of Rh1 to Rg2 by synPgURT94 and synPgURT94m1, respectively, then why authors did not find rh1 in the analysis. What about the remaining (unused) rh1 substrate?

The authors tried to explain these (lines 200-209) but unable to understand what was the substrate and other conditions used here?

Line 111-112- PgURT94 showed expression in other parts also. Is it the author's qPCR data? Or it's from the database?

Line 132-140: Sugar donor specificity of PgURT94. Details are missing in the methods. Only TLC analysis?

The manuscript requires careful editing and re-write in a logical manner to help readers to follow this manuscript

Reviewers' comments:

Reviewer #1 (Remarks to the Author):

The authors have successfully attempted to engineer the bio synthetic pathway for production of rhamnosylated ginsenosides in a yeast chassis based production module in a high titre.

Indeed, information related to the rhamnosyl transferases particularly were extremely limited and this report adds tremendously to our present understanding of the biosynthetic pathway of the panaxosides. The work is technically sound and should be easily reproducible. The narrative is well written, concise and is easy to interpret. The results are well explained and readily comprehensible.

I recommend the manuscript to be accepted in the present form. However I would like the authors to comment on the ginsenoside extraction protocol that they have employed from fermentation broth. The ginsenosides were extracted directly from the organic upper layer of n butanol and analyzed further. However, this may also result in loss of ginsenosides during the phase separation. Have the authors tested for ginsenoside recovery using this extraction method and estimated a percentage recovery?

Since extraction of ginsenosides vary with solvents and methanol is almost universally used for ginsenoside extraction. In case of broths, they may be lyophilized and then extracted using methanol and then phase separation using n butanol may be done. Please refer to the following for more clarity on my query.

Biswas T, Ajayakumar PV, Mathur AK, Mathur A. Solvent-based extraction optimisation for efficient ultrasonication-assisted ginsenoside recovery from *Panax quinquefolius* and *P. sikkimensis* cell suspension lines. *Natural Product Research*. 2015 Jul 3;29(13):1256-63.

>Response:

Thanks a lot for your positive comments on our manuscript. According to your suggestions, we have now performed an additional experiment to evaluate the recovery rate of the current extraction protocol using *n*-butanol. Using standard samples of Rg2

and Re aqueous solutions, direct extraction using *n*-butanol reached a recovery rate of 92.3% and 90.1%, respectively (Figure R1).

Figure R1 Ginsenosides Rg2 and Re recovery rate assay by *n*-butanol extraction method.

We really appreciate the extraction method you suggested for us, which is supported by a wonderful paper published in Natural Product Research. We realized that methanol is a great extraction agent and we have no doubts that the method you provide will ensure an improvement in extraction efficiency. However, we think a “lyophilizing–methanol extracting–*n*-butanol phase separation” process is much complicated than direct extraction using *n*-butanol, which might not be suitable for industrialized application. Since direct extraction using *n*-butanol could reach a recovery rate of more than 90% as mentioned above, we thus demined to use this method.

I would like to congratulate the authors for this successful attempt to unravel the complex biosynthetic pathways for bioactive panaxosides and opening up avenues for their large scale production via synthetic biology.

>**Response:**

Thanks a lot for your constructive suggestions and encouragements.

Reviewer #2 (Remarks to the Author):

Ginsenosides showed various pharmacological activities that have been wide used for preventing cardiovascular and cerebrovascular diseases. Rg2 and Re are main ginsenosides with sugar moiety in Panax plants root which take 3-4 years to harvest. The manuscript elucidated the biosynthesis pathway of these two ginsenosides and high yield production in engineered *Saccharomyces cerevisiae* with synthetic biology strategy. The products and yield were proved by TLC, NMR and LCMS which were persuasive enough to me. This work is well fundamental build and potential to produce large-scale ginsenosides in days.

>Response:

Thanks a lot for your positive comments and constructive suggestions.

In Suppl. Fig.5. The chemistry structures need to be redrawn.

>Response:

Thanks. We have now revised the Suppl. Fig.5 (Page 9 in **Supporting Information**).

Reviewer #3 (Remarks to the Author):

The manuscript describes continuing research of the authors on the rhamnosylated ginsenosides pathway in Panax ginseng and their production by engineered *Saccharomyces cerevisiae*. The study is promising, and the paper contains some interesting results of potential scientific significance. However, there is major insufficiency that the investigators need to address in the organization and presentation of the paper. It must be revised in a logical flow and clear manner in order to further bring out the objectives and importance of the various experiments. The figure legends are not properly explained, it should be in detail. need consistency in the use of gene, vector/strain names (e.g. PPT, pET28a-RH1, pgURT94-Rh1), and repetition of the same information e.g. introduction: lines 61-61, lines 82-83, and lines 96-98 convey the same message.

>Response:

Thanks a lot for your positive comments and constructive suggestion to improve our manuscript. In order to response to your concerns more clearly, we listed our point-to-point reply below:

1) However, there is major insufficiency that the investigators need to address in the organization and presentation of the paper. It must be revised in a logical flow and clear manner in order to further bring out the objectives and importance of the various experiments.

Thanks a lot for this constructive suggestion. In the revised version, we have carefully checked and revised the whole manuscript to improve its organization and presentation. In particular, we have revised the **Results and Discussion** section and make sure the organization of each experiment are in a logical flow, reasons and aims for each experiment has been emphasized. We have also introduced the limitation of traditional ginsenosides Rg2 and Re manufacturing methods in the **Introduction** section to highlight the objectives and importance our study (line 43-48).

2) The figure legends are not properly explained, it should be in detail.

Thanks a lot, we have revised each figure legend carefully to make sure the data has been properly explained (line 566-615).

3) Need consistency in the use of gene, vector/strain names (e.g. PPT, pET28a-RH1, pgURT94-Rh1).

Thanks, we have checked and revised the manuscript to make sure the consistency in the use of gene, enzyme, vector, strain names. Generally, the names of **enzymes** were uniformly written as standard names (e.g. PgURT94, PgDDS, CYP716A47); the names of **genes** were written as their corresponding enzymes' in italics (e.g. *PgURT94*, *PgDDS*, *CYP716A47*); **vector/plasmid's** names were presented as "commercial vector name-the harboring genes" (e.g. pET28a-*PgURT94*); yeast **strains'** name were written as "the target product-number" (e.g. PPT-10, Rg2-04, Re-01).

4) Repetition of the same information e.g. introduction: lines 61-61, lines 82-83, and lines 96-98 convey the same message.

Thanks, we have revised the repetitive description you mentioned (Line 68-70, Line 89-90 and 102-103) and checked whole manuscript to avoid similar mistakes.

The methodology part needs to improve with more clarity: substrate and its concentration used and other details for each experiment should be mentioned.

>**Response:**

Thanks a lot for this constructive suggestion, we have carefully checked and revised the methodology part. We have tried our best to make this part more clarity and provide the information about substrate, concentration, incubation/reaction time, reaction volume/temperature/pH and other details for each experiment in **Methods** section.

Figure 4 shows only the peaks, which are already shown in the figures before. Better to bring some meaningful data from supplementary.

>**Response:**

Thanks a lot for your suggestion, we have now revised Figure 4. The conversion ratio data were provided as Figure 4c & d in a bar graph form, which were from supplementary table S7.

In figure 2 a and figure 4a authors find peaks for rh1 but in other places they did not find it.

Table S7 and lines 192-193 indicate 70.6% and 92.4% conversion of Rh1 to Rg2 by synPgURT94 and synPgURT94m1, respectively, then why authors did not find rh1 in the analysis. What about the remaining (unused) rh1 substrate?

The authors tried to explain these (lines 200-209) but unable to understand what was the substrate and other conditions used here?

>**Response:**

Sorry for the confusion here. The data of Figure 2a, Figure 4a and Table S7 were all from *in vitro* enzyme catalysis, which were conducted by incubating Rh1 substrate with crude enzymes of PgURT94 or its mutant. Just as you point out, peaks for Rh1 are clearly visible in Figure 2a and Figure 4a, which indicated the remaining (unused) Rh1 substrate owing to their <100% substrate conversions to Rh1 (70.6% and 92.4% conversion of Rh1 to Rg2 by synPgURT94 and synPgURT94m1, respectively).

While in other places including Figure 3, Figure 5, and Figure 6, they are all *in vivo* yeast cell factories data, which were conducted by reconstruction the complete biosynthetic pathway of Rg2 or Re in yeast strain. In these cases, Rg2 or Re were synthesized directly from glucose (no Rh1 substrate was added). Actually, Rh1 (as a metabolic intermediate) did not accumulate in any of the engineered yeast strains, that's why we did not find Rh1 peaks in these assays.

We speculated the conversion of PPT to Rh1 might be a limiting step during the complete biosynthetic pathway of Rg2 and the higher conversion of Rh1 to Rg2 by synPgURT94 and synPgURT94m1 in yeast. Thus, in lines 200-209, we were trying to address this problem by introducing an additional copy of PgUGT71A54.

Line 111-112- PgURT94 showed expression in other parts also. Is it the author's qPCR data? Or it's from the database?

>**Response:**

Thanks, the expression pattern analysis of *PgURT94* was based on the *Panax ginseng* transcriptome data from public database. These data were listed in Supplementary Table S10 and S11 (Page 21-22 in **Supporting Information**).

Line 132-140: Sugar donor specificity of PgURT94. Details are missing in the methods. Only TLC analysis?

>**Response:**

Thanks a lot for pointing out our omission here. Previously, we only used TLC analysis to determine the sugar donor specificity of PgURT94. Here, according to you suggestion, we have now added HPLC analysis data of this assay (**Supplementary Fig. S4**). Besides, we have revised the manuscript and added the detailed methods of this assay (Line 340-344).

The manuscript requires careful editing and re-write in a logical manner to help readers to follow this manuscript

>**Response:**

Thanks a lot for this suggestion, we have carefully revised the whole manuscript and tried our best to improve its description, organization, and logicity.

We would like to take this opportunity to thank you again for your supportive and helpful guidance in how to improve our study.

REVIEWERS' COMMENTS:

Reviewer #1 (Remarks to the Author):

paper is recommended for acceptance.

Reviewer #3 (Remarks to the Author):

The authors have extensively revised and improved the manuscript.

I would suggest a few minor points that authors may think of

1. There is a scope for improvement of the abstract. Please add more results and highlight the novel output of the study rather than the methods.
2. Paragraph (line 226-235): please add the source of data (Table/figure).

Reviewers' comments:

Reviewer #1 (Remarks to the Author):

paper is recommended for acceptance.

>Response:

Thanks a lot for your positive comments on our manuscript.

Reviewer #3 (Remarks to the Author):

The authors have extensively revised and improved the manuscript.

I would suggest a few minor points that authors may think of

1. There is a scope for improvement of the abstract. Please add more results and highlight the novel output of the study rather than the methods.

>Response:

Thanks a lot for your positive comments and constructive suggestions. We have revised the abstract according to your suggestion, methods have been simplified and the background and output of the study have been highlighted.

2. Paragraph (line 226-235): please add the source of data (Table/figure).

>Response:

Thanks a lot for your suggestion, we have revised the manuscript and added the related source of data (Line 212-215).